# Risk-stratified treatment for drug-susceptible pulmonary tuberculosis

Vincent K. Chang[1,2], Marjorie Z. Imperial[1,2], Patrick P. J. Phillips [2,3], Gustavo E. Velásquez[2,4], Payam Nahid[2,3], Andrew Vernon[5], Ekaterina V. Kurbatova[5], Susan Swindells[6], Richard E. Chaisson [7], Susan E. Dorman[8], John L. Johnson [9,10], Marc Weiner[11], Amina Jindani[12], Thomas Harrison[12], Erin E. Sizemore[5], William Whitworth[5], Wendy Carr[5], Kia E. Bryant[5], Deron Burton[5], Kelly E. Dooley[13], Melissa Engle[11], Pheona Nsubuga[10], Andreas H. Diacon[14], Nguyen Viet Nhung[15,16], Rodney Dawson[17], Radojka M. Savic [1,2] ✉, AIDS Clinical Trial Group* & Tuberculosis Trials Consortium*

The Phase 3 randomized controlled trial, TBTC Study 31/ACTG A5349 (NCT02410772) demonstrated that a 4-month rifapentine-moxifloxacin regimen for drug-susceptible pulmonary tuberculosis was safe and effective. The primary efficacy outcome was 12-month tuberculosis disease free survival, while the primary safety outcome was the proportion of grade 3 or higher adverse events during the treatment period. We conducted an analysis of demographic, clinical, microbiologic, radiographic, and pharmacokinetic data and identified risk factors for unfavorable outcomes and adverse events. Among participants receiving the rifapentine-moxifloxacin regimen, low rifapentine exposure is the strongest driver of tuberculosis-related unfavorable outcomes (HR 0.65 for every 100 μg•h/mL increase, 95%CI 0.54–0.77). The only other risk factors identified are markers of higher baseline disease severity, namely Xpert MTB/RIF cycle threshold and extent of disease on baseline chest radiography (Xpert: HR 1.43 for every 3-cycle-threshold decrease, 95%CI 1.07–1.91; extensive disease: HR 2.02, 95%CI 1.07–3.82). From these risk factors, we developed a simple risk stratification to classify disease phenotypes as easier-, moderately-harder, or harder-to-treat TB. Notably, high rifapentine exposures are not associated with any predefined adverse safety outcomes. Our results suggest that the easier-to-treat subgroup may be eligible for further treatment shortening while the harder-to-treat subgroup may need higher doses or longer treatment.

TBTC Study 31/A5349 was a Phase 3 international multicenter randomized controlled trial that compared 4-month regimens of daily isoniazid, rifapentine, and pyrazinamide plus either moxifloxacin (rifapentine-moxifloxacin regimen) or ethambutol (rifapentine-regimen) to the 6-month standard treatment of isoniazid, rifampin, pyrazinamide, and ethambutol (control regimen) for the treatment of

drug-susceptible pulmonary tuberculosis. The 4-month rifapentine-moxifloxacin regimen demonstrated noninferior efficacy and comparable safety to the control (primary results published in NEJM)[1,2], making it the first 4-month regimen endorsed by both the World Health Organization and the U.S. Centers for Disease Control and Prevention (CDC) for the treatment of adolescents and adults with

---

A full list of affiliations appears at the end of the paper. *Lists of authors and their affiliations appears at the end of the paper. ✉e-mail: rada.savic@ucsf.edu

pulmonary tuberculosis[3–5]. While the rifapentine-regimen was not shown to have noninferior efficacy to the control, 82% of participants receiving it were cured[2].

Tuberculosis has long been treated with a one-size-fits-all 6-month regimen, as in the control regimen of Study 31/A5349. However, there is increasing evidence that a subset of patients are overtreated, and those with harder-to-treat TB (smear grade 3+ and cavitary disease) require longer than the prescribed treatment duration[6–9]. Similarly, it has been shown that the suboptimal efficacy of experimental regimens containing rifamycins and fluoroquinolones for persons with harder-to-treat TB is the primary reason underlying the unfavorable clinical outcomes in recent Phase 3 clinical trials[6]. It is therefore essential that we understand the key drivers of treatment response in disease phenotypes to help define the utility of current regimens and best practices for late-stage tuberculosis drug trials. To that end, Study 31/A5349 incorporated pharmacokinetic sampling for all antituberculosis drugs among all participants, providing an unprecedented opportunity to establish the contribution of exposure-response relationships to clinical outcomes for antituberculosis drugs, and permitting insights into the complex interplay between disease severity, participant characteristics, regimen potency, and regimen duration on long-term clinical outcomes[1].

Here, we report the results of prespecified Study 31/A5349 secondary analyses designed to assess pharmacokinetic, clinical, and demographic markers for efficacy and safety outcomes among participants treated with the 4-month rifapentine or rifapentine-moxifloxacin regimens. Our objectives were to define risk-stratified approaches for the optimal use of the novel 4-month rifapentine-moxifloxacin regimen in clinical practice, to provide evidence for why the 4-month rifapentine-regimen did not meet the noninferiority margin, and to define disease phenotypes that were successfully cured with the 4-month rifapentine-regimen.

## Results

The microbiologically eligible population consisted of 2343 participants, of which 768 were in the control group, 784 in the rifapentine group, and 791 in the rifapentine-moxifloxacin group. The study population was mostly male (71%) and of self-reported Black race (72%), with 11% self-reporting Asian race and 15% mixed race; the median age was 30 years, and 8% of the study population was living with HIV. Other participant characteristics are reported in Table 1. Trial-level Kaplan Meier estimates by regimen are shown in Fig. S1.

### Risk factors for tuberculosis-related unfavorable outcomes

Stratified Kaplan-Meier estimates and univariate Cox regression analysis (Fig. S2) demonstrated that below median exposures were associated with increased hazard of tuberculosis-related unfavorable outcomes for rifapentine (rifapentine-regimen: HR 3.81, 95% CI 2.22–6.55; rifapentine-moxifloxacin: HR 2.23, 95% CI 1.18–4.20), moxifloxacin (HR 2.00, 95% CI 1.09–3.65), isoniazid (rifapentine-regimen: HR 1.79, 95% CI 1.09–2.95; not significant in control or rifapentine-moxifloxacin regimens), ethambutol (rifapentine-regimen: HR 1.73, 95% CI 1.10–2.72; control: HR 2.43, 95% CI 1.01–5.88), and pyrazinamide (not significant in rifapentine-regimen; rifapentine-moxifloxacin: HR 1.85, 95% CI 1.03–3.33; control: HR 2.46, 95% CI 1.10–5.54). Low rifampin exposures were not associated with an increased hazard in the control arm.

Participants with missing data were excluded from multivariable analyses. Participants included: rifapentine-moxifloxacin regimen: 688 of 791 participants (87%), rifapentine-regimen: 675 of 784 participants (86%), control regimen: 667 of 768 participants (87%). Among participants receiving rifapentine-moxifloxacin, extensive disease on chest radiography (defined as involvement of ≥50% thoracic cavity area on chest radiography) and lower baseline Xpert MTB/RIF cycle threshold were associated with an increased hazard of tuberculosis-related

unfavorable outcomes (extensive disease: HR 2.02, 95% CI 1.07–3.82; Xpert: HR 1.43 for every 3–cycle-threshold decrease, 95% CI 1.07–1.91). Higher rifapentine $AUC_{0-24h}$ was associated with a decreased hazard (HR 0.65 for every 100 µg•h/mL increase, 95% CI 0.54–0.77). All other effects did not meet statistical criteria for inclusion in the model after adjusting for rifapentine exposure (Fig. 1a).

Among participants receiving the rifapentine-regimen, older age, lower weight, and lower baseline Xpert MTB/RIF cycle threshold were associated with an increased hazard of tuberculosis-related unfavorable outcomes (age: HR 1.38 for every 10-year increase, 95% CI 1.13–1.68; weight: HR 1.76 for every 10-kg decrease, 95% CI 1.25–2.49; Xpert: HR 1.54 for every 3–cycle-threshold decrease, 95% CI 1.24–1.93). Higher rifapentine $AUC_{0-24h}$ was associated with a decreased hazard (HR 0.77 for every 100 µg•h/mL increase, 95% CI 0.63–0.95). Extensive disease was associated with increased hazard in the baseline-factors-only multivariable model (Fig. S3) but not after adjusting for rifapentine exposure (HR 1.61 with ≥50% thoracic cavity, 95% CI 0.98–2.65) (Fig. 1b).

Among participants assigned to the control regimen, lower baseline Xpert MTB/RIF cycle threshold was associated with an increased hazard of tuberculosis-related unfavorable outcomes, and higher pyrazinamide $AUC_{0-24h}$ (but not rifampin) was associated with a decreased hazard (Xpert MTB/RIF: HR 1.69 for every 3–cycle-threshold decrease, 95% CI 1.08–2.63; Pyrazinamide: HR 0.35 for every 100 µg•h/mL increase, 95% CI 0.15–0.83) (Fig. 1c).

Cox proportional hazards assumption were assessed in Tables S1–S3 and demonstrated that most covariates met the proportional hazards assumption. Univariate Cox analysis and univariate subgroup analyses can be found in the supplement which yielded similar results to the primary multivariable analysis (Tables S4-S6, Fig. S4). Sensitivity analysis excluding imputed PK values demonstrated limited impact of population PK model imputation of missing PK values on the primary analysis (Tables S7, S8).

### Risk Stratification Algorithm

We designed a simple risk algorithm for participants receiving the rifapentine-moxifloxacin regimen: Xpert MTB/RIF cycle threshold stratified above and below the median (17.3 rounded to 18, the median cycle threshold value for participants with smear grade 1+, Fig. S5) and extent of disease on chest radiography (above and below 50% involvement of thoracic area). Rifapentine exposure was excluded from the risk algorithm as it requires therapeutic drug monitoring and thus is not always available to clinicians in programmatic settings, and is not available at baseline given the three-week autoinduction period of rifamycins[10]. Easier-to-treat TB was defined as Xpert MTB/RIF cycle threshold ≥18 and involvement of <50% thoracic area, harder-to-treat TB was defined as Xpert MTB/RIF cycle threshold <18 and involvement of ≥50% thoracic area, while the remaining population with either (i) Xpert MTB/RIF cycle threshold <18 and involvement of <50% thoracic area or (ii) Xpert MTB/RIF cycle threshold ≥18 and involvement of ≥50% thoracic area was defined as moderately-harder-to-treat TB. Kaplan-Meier estimates stratified by regimen, disease phenotype, and rifapentine exposure demonstrated that among participants with above-median rifapentine exposure, 12-month outcomes were comparable across arms. In contrast, in participants with below-median rifapentine exposure, the substitution of moxifloxacin for ethambutol improved 12-month outcomes across all risk groups (low-risk: 6.6% to 4.4%; moderate-risk: 11.3% to 6.1%; high-risk: 29.4% to 14.3%) (Fig. 2).

### Disease phenotype subgroup analyses

These disease phenotypes demonstrated similar rates of tuberculosis-related unfavorable outcomes across the rifapentine-moxifloxacin and control regimens in the subpopulations at low risk (risk difference 0.1%, 95% CI−3.4%–3.6%) and moderate risk (risk

**Table 1 | Summary of Demographics, Clinical Factors, Pharmacokinetics, Treatment and Safety Outcomes in the Micro-biologically Eligible Population from Study 31/A5349**

| | Rifapentine-Moxifloxacin 2HPZM/2HPM | Rifapentine 2HPZE/2HP | Control 2HRZE/4HR | Missing |
|---|---|---|---|---|
| Number of Participants | 791 | 784 | 768 | – |
| DEMOGRAPHIC FACTORS | | | | |
| Age [years] | 31 (17–60) | 30 (18–59) | 30 (18–60) | 0 (0) |
| Male Sex | 563 (71) | 563 (72) | 544 (71) | 0 (0) |
| Weight [kg] | 53 (41–76) | 53 (41–75) | 53 (41–75) | 1 (0) |
| BMI [kg/m2] | 19.03 (15.23–27.90) | 18.92 (14.87–27.54) | 18.93 (15.03–27.37) | 1 (0) |
| Race | | | | 0 (0) |
| Black | 552 (70) | 571 (73) | 553 (72) | |
| Mixed/Multiracial | 137 (17) | 112 (14) | 114 (15) | |
| Asian | 89 (11) | 93 (12) | 86 (11) | |
| White | 13 (2) | 8 (1) | 15 (2) | |
| African Clinical Site | 578 (73) | 573 (73) | 565 (74) | 0 (0) |
| BASELINE CLINICAL FACTORS | | | | |
| Xpert MTB/RIF cycle threshold | 17.2 (10.6–24.3) | 17.4 (11.8–25.7) | 17.2 (11.5–25.2) | 305 (13) |
| Time to Detection on Sputum Liquid Culture [days] | 8.12 (3.10–3.17) | 7.92 (3.82–19.01) | 8.21 (3.73–19.3) | 65 (3) |
| Cavitary Disease on Chest Radiography | 572 (72) | 572 (73) | 557 (73) | 0 (0) |
| Aggregate Cavity Size on Chest Radiography | | | | 15 (0.7) |
| No cavities | 213 (27) | 206 (26) | 206 (27) | |
| Cavities <4 cm | 277 (35) | 246 (32) | 251 (33) | |
| Cavities ≥ 4 cm | 295 (38) | 327 (42) | 307 (40) | |
| Extent of Disease on Chest Radiography | | | | 15 (0.7) |
| Lesions (<25%) thoracic area | 155 (20) | 135 (17) | 120 (16) | |
| Lesions (25% to <50%) thoracic area | 360 (46) | 343 (44) | 343 (45) | |
| Lesions (≥50%)thoracic area | 270 (34) | 301 (39) | 301 (39) | |
| Sputum AFB Smear Grade | | | | 2 (0.1) |
| Negative | 29 (4) | 32 (4) | 21 (3) | |
| Scanty | 149 (19) | 127 (16) | 121 (16) | |
| Grade 1 | 168 (21) | 173 (22) | 188 (25) | |
| Grade 2 | 228 (29) | 228 (29) | 229 (30) | |
| Grade 3 | 209 (26) | 214 (27) | 198 (26) | |
| Positive (WHO scale not used) | 7 (1) | 9 (1) | 10 (1) | |
| Karnofsky Score | 90 (70–100) | 90 (70–100) | 90 (70–100) | 0 (0) |
| Living with HIV | 62 (8) | 68 (9) | 64 (8) | 1 (0) |
| CD4 Count | 350 (118–673) | 346 (133–795) | 334 (108–773) | 0 (0) |
| History of Diabetes | 32 (4) | 14 (2) | 31 (4) | 0 (0) |
| Smoking History | | | | 0 (0) |
| Never | 431 (54) | 409 (52) | 391 (51) | |
| Current | 185 (23) | 175 (22) | 181 (24) | |
| Former | 175 (22) | 200 (26) | 196 (26) | |
| History of Liver Disease | 6 (1) | 6 (1) | 5 (1) | 0 (0) |
| Prior Episode of TB | 97 (12) | 85 (11) | 83 (11) | 0 (0) |
| Time since Prior Episode of TB [years] | 7.8 (0.9–42.7) | 6.2 (0.9–33.6) | 7.6 (0.7–40.6) | 0 (0) |
| PHARMACOKINETICS | | | | |
| Rifapentine $AUC_{0-24h}$ [µg•h/mL] | 557.1 (276–983) | 562.4 (302–1037) | – | 67 (4) |
| Moxifloxacin $AUC_{0-24h}$ [µg•h/mL] | 24.3 (15.3–44.1) | – | – | 49 (6) |
| Isoniazid $AUC_{0-24h}$ [µg•h/mL] | 8.4 (6.0–24.6) | 8.4 (6.0–23.1) | 10.8 (7.4–28.8) | 153 (7) |
| Pyrazinamide $AUC_{0-24h}$ [µg•h/mL] | 350 (229–590) | 307 (214–553) | 353 (258–577) | 125 (5) |
| Ethambutol $AUC_{0-24h}$ [µg•h/mL] | – | 15.7 (12.4–21.1) | 15.0 (11.8–20.5) | 138 (9) |
| Rifampin $AUC_{0-24h}$ [µg•h/mL] | – | – | 41.4 (22.1–147.1) | 57 (6) |
| Rifapentine $C_{max}$ [µg /mL] | 32.8 (19.2–51.9) | 32.4 (19.5–53.4) | – | 67 (4) |
| Moxifloxacin $C_{max}$ [µg /mL] | 2.56 (1.72–.35) | – | – | 49 (6) |
| Isoniazid $C_{max}$ [µg /mL] | 2.1 (1.2–3.1) | 2.0 (1.2–2.7) | 2.6 (1.9–3.2) | 153 (7) |
| Pyrazinamide $C_{max}$ [µg /mL] | 28.6 (19.0–44.8) | 28.1 (22.3–42.8) | 33.8 (26.9–46.6) | 125 (5) |

**Table 1 (continued) | Summary of Demographics, Clinical Factors, Pharmacokinetics, Treatment and Safety Outcomes in the Microbiologically Eligible Population from Study 31/A5349**

| | Rifapentine-Moxifloxacin 2HPZM/2HPM | Rifapentine 2HPZE/2HP | Control 2HRZE/4HR | Missing |
|---|---|---|---|---|
| Ethambutol $C_{max}$ [μg /mL] | – | 1.68 (1.05–3.14) | 1.83 (1.24–3.12) | 138 (9) |
| Rifampin $C_{max}$ [μg /mL] | – | – | 8.6 (4.8–22.9) | 57 (6) |
| ADHERENCE | | | | |
| Participants who received 95% of planned doses | 734 (93) | 743 (95) | 705 (92) | 7 (0) |
| TREATMENT OUTCOMES | | | | |
| Tuberculosis-Related Unfavorable Outcomes | 45 (5.7) | 75 (9.5) | 24 (3.1) | – |
| Not Tuberculosis-Related Unfavorable Outcomes | 43 (5.4) | 32 (4.1) | 46 (5.9) | – |
| Total Unfavorable Outcomes | 88 (11.1) | 107 (13.6) | 70 (9.1) | – |
| SAFETY OUTCOMES | | | | |
| Number of Participants (Safety Population) | 846 | 835 | 825 | – |
| Grade 3–5 adverse event | 159 (18.8) | 119 (14.3) | 159 (19.3) | – |
| Treatment-related grade 3–5 adverse event | 109 (12.9) | 64 (7.7) | 81 (9.8) | – |
| Any serious adverse event | 37 (4.4) | 39 (4.7) | 56 (6.8) | – |
| Death | 3 (0.4) | 4 (0.5) | 7 (0.8) | – |
| Premature discontinuation of assigned regimen for any reason in the microbiologically eligible population | 54/791 (6.8) | 37/784 (4.7) | 61/768 (7.9) | – |

The microbiologically eligible population excluded randomized participants for drug resistance, *M. tuberculosis*-negative culture, or violation of eligibility criteria at baseline. Data are shown as *n* (%) for categorical measures and median (2.5th and 97.5th percentiles) for continuous measures.Abbreviations: AFB, acid-fast bacillus; $AUC_{0-24h}$, area under the concentration-time curve from 0–24 hours; BMI, body mass index; $C_{max}$, maximal plasma concentration; HIV, human immunodeficiency virus

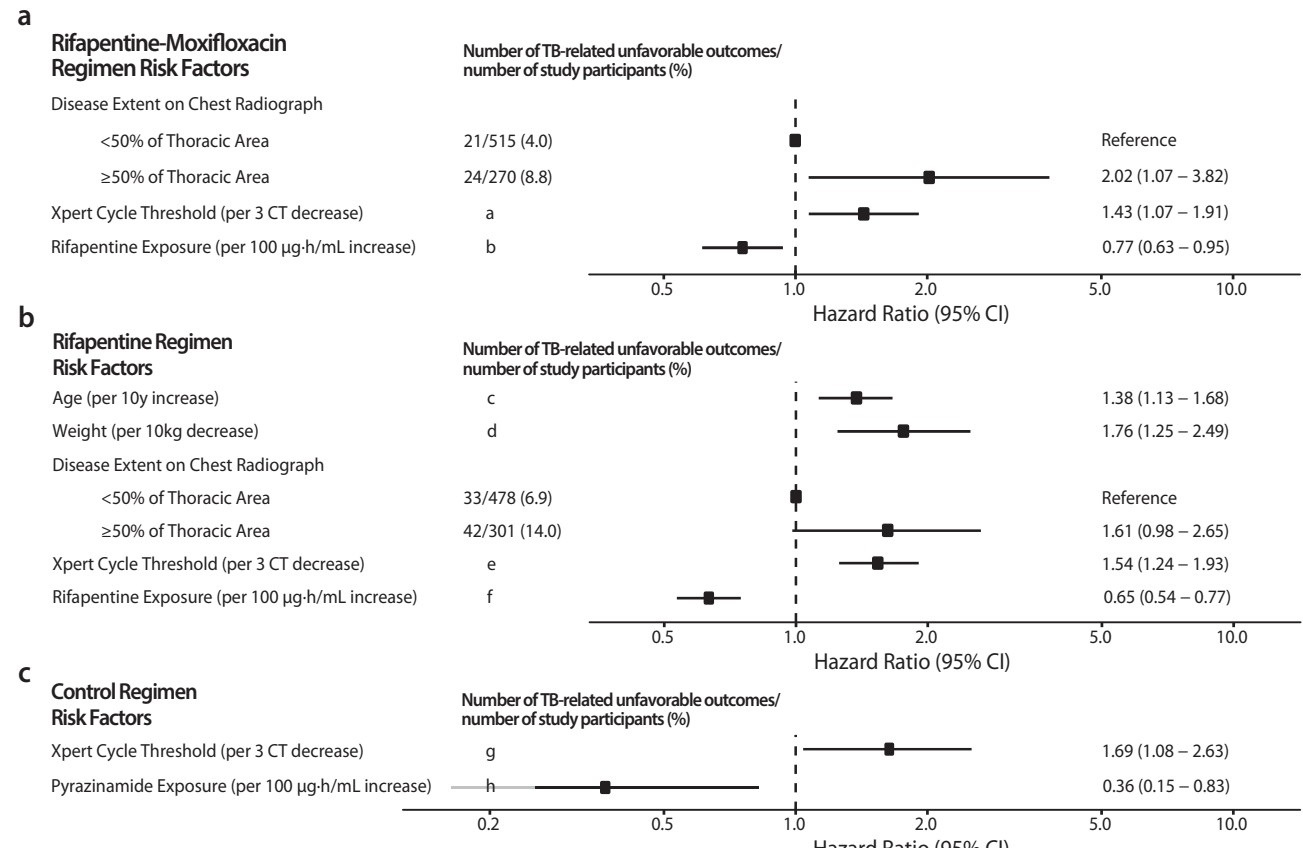

**Fig. 1 | Multivariable Hazard Ratios for Tuberculosis-Related Unfavorable Outcomes.** Multivariable analysis of pharmacokinetic and baseline predictors for **a** rifapentine-moxifloxacin, **b** rifapentine, and **c** control regimens. Data are presented as hazard ratio estimates (point) and 95% confidence intervals (error bars). **a** Xpert MTB/RIF cycle threshold <18, 29/397 (7.3); Xpert MTB/RIF cycle threshold ≥ 18, 10/296 (3.4), **b** Rifapentine exposure <560 μg·h/mL, 31/402 (7.7); Rifapentine exposure ≥ 560 μg·h/mL, 14/389 (3.6), **c** Age <30 years, 21/354 (5.9); Age ≥ 30 years, 54/430 (12.6), **d** Weight <53 kg, 45/364 (12.4); Weight ≥ 53 kg, 30/419 (7.2), **e** Xpert MTB/RIF cycle threshold <18, 54/397 (13.6); Xpert MTB/RIF cycle threshold ≥ 18, 13/284 (7.7), **f** Rifapentine exposure <560 μg·h/mL, 58/386 (15.0); Rifapentine exposure ≥ 560 μg·h/mL, 17/398 (4.3), **g** Xpert MTB/RIF cycle threshold <18, 15/399 (3.7); Xpert MTB/RIF cycle threshold ≥ 18, 5/268 (1.9), **h** Pyrazinamide exposure <336 μg/mL, 14/304 (4.6); Pyrazinamide exposure ≥ 336 μg/mL, 10/462 (2.2).

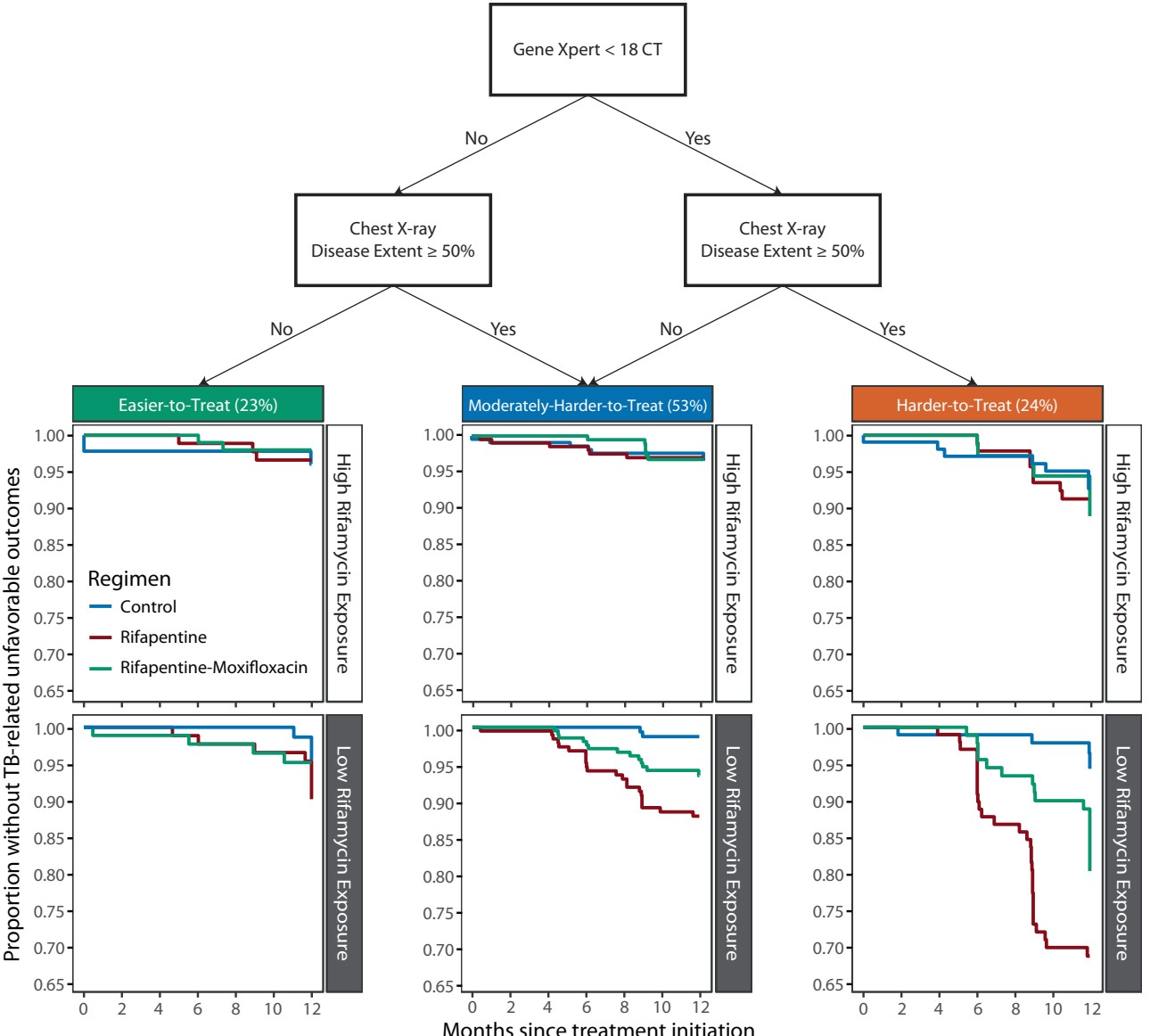

**Fig. 2 | Xpert MTB/RIF cycle threshold and extent of disease on chest radiography stratify participants into easier-to-treat TB, moderately-harder-to-treat TB, and harder-to-treat TB disease phenotypes.** Disease phenotypes were defined by baseline Xpert MTB/RIF cycle threshold and extent of disease on chest radiography, defined as the percent involvement of the area of the thoracic cavity. Disease phenotypes were further stratified by rifamycin exposure, where Kaplan Meier estimates demonstrated that easier-to-treat TB does not need exposure optimization. Moderately-harder-to-treat TB among participants receiving the rifapentine-regimen would require dose optimization to achieve optimal outcomes. Participants with moderately-harder-to-treat TB receiving the rifapentine-moxifloxacin regimen would benefit from dose optimization, however this would not be required to achieve optimal outcomes. Participants with harder-to-treat TB and high rifamycin exposure have similar outcomes across regimens, but none of the regimens achieve <5% tuberculosis-related unfavorable outcomes regardless of rifamycin exposure levels.

difference 2.5%, 95% CI 0.1–4.9%). High-risk participants had higher rates than control (risk difference 6.2%, 95% CI 0.5–11.9%). (Fig. 3a) Among participants receiving the rifapentine regimen, we observed similar rates of tuberculosis-related unfavorable outcomes when compared to control in the subpopulations at low risk (risk difference 2.1%, 95% CI−2.0–6.1%). However, those classified as moderate- or high-risk experienced higher rates than control (moderate-risk: risk difference 4.8%; 95% CI 2.0–7.7%; high-risk: risk difference 13.9%; 95% CI, 7.6–20.2%). (Fig. 3b) TB-ReFLECT disease phenotypes (harder-to-treat: Smear grade 2+ and cavitary disease) previously described by Imperial et al. did not show differences in TB-related unfavorable outcomes for participants on control and rifapentine-moxifloxacin regimens but did for participants on the rifapentine regimen (Fig. S6) [6].

## Impact of adherence

Low numbers of participants were non-adherent, 705/768 (92%), 743/784 (95%), and 734/791 (93%) participants were administered 95% of planned doses in the control, rifapentine, and rifapentine-moxifloxacin regimens respectively. In univariate analysis, adherence was associated with increased hazard of TB-related unfavorable outcome in all regimens (rifapentine-moxifloxacin: HR 1.22 for every week of missed doses, 95% CI 1.11–1.33; rifapentine regimen: HR 1.16, 95% CI 1.05–1.28; control: HR 1.45, 95% CI 1.23–1.72) (Fig. S7). In multivariable analysis, adherence was associated with increased hazard of TB-related unfavorable outcome in the rifapentine-moxifloxacin regimen (HR 1.31 for every week of missed doses, 95% CI 1.19–1.44) and control (HR 1.37, 95% CI 1.12–1.67), but did not show a significant association in the rifapentine regimen (HR 1.10, 95% CI 0.96–1.27) (Fig. 1, Table S9).

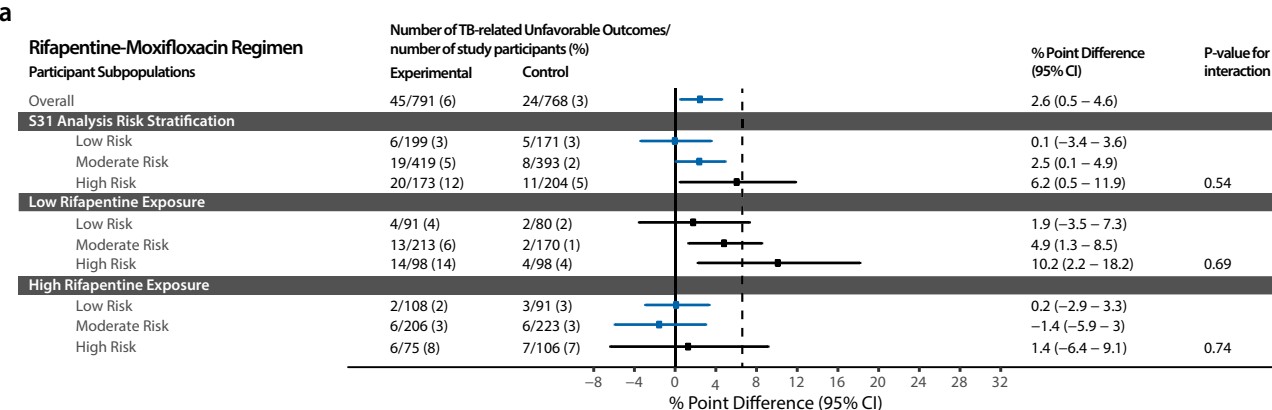

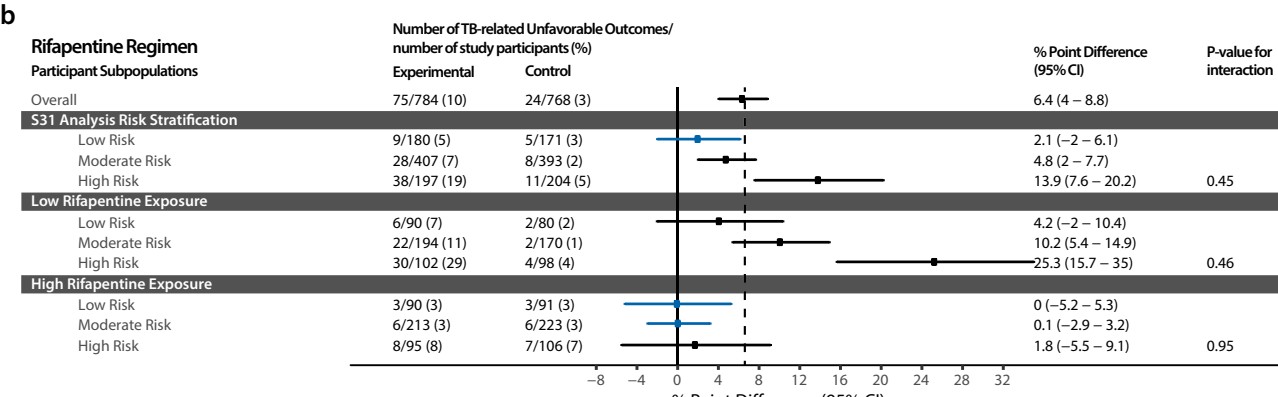

**Fig. 3 | Risk Stratification Reveals a Low-Risk Subgroup where Further Treatment Shortening and Simplification is Likely Possible and a High-Risk Subgroup where Longer Treatment May Be Needed.** The figure shows the results of subgroup analyses of Study 31/A5349 risk groups, data are presented as percentage point differences (point) and 95% confidence intervals (error bars). Low and high rifapentine subgroups in the experimental arms were compared to low and high rifampin subgroups in the control arm. Two-tailed interaction *p*-values tested for interaction between regimen (experimental vs. control) and the disease phenotypes in a Cox proportional hazards model. **a** Analysis of the rifapentine-moxifloxacin regimen demonstrates that the high-risk group, comprising 23% of the Study 31/A5349 population, may require a longer and/or more potent regimen to achieve ≤ 5% unfavorable outcomes. **b** Analysis of the rifapentine-regimen demonstrates that the subpopulations at low risk regardless of rifapentine exposure, and moderate- or high-risk with high rifapentine exposure, comprising 62% of the Study 31/A5349 population in the rifapentine arm, have small differences in outcome when compared to the control. Additionally, in both rifapentine and rifapentine-moxifloxacin regimens among participants with high rifapentine exposure, the percentage point differences between experimental and control regimens are small (<1.8%) across all risk groups.

## Safety

Grade 3–5 adverse events by regimen are reported in Table S10. In participants receiving rifapentine-moxifloxacin regimens, higher pyrazinamide exposures were associated with increased risk of any grade 3–5 adverse events (OR 1.22 for every 100 μg•h/mL increase in $AUC_{0-24h}$, 95% CI 1.02–1.45) and treatment-related grade 3–5 adverse events (OR 1.27, 95% CI 1.04–1.55). There were, however, no significant associations between continuous rifapentine exposure and any of the five composite safety outcomes. (Fig. 4). In univariate analysis, older age, decreasing Karnofsky performance score at baseline, and history of liver disease were also associated with any grade 3–5 adverse events (Table S11). In multivariable analysis, older age (OR 1.22 for every 10-year increase, 95% CI 1.06–1.41), history of liver disease (OR 7.43, 95% CI 1.42–54.3), and higher pyrazinamide exposures (OR 1.23, 95% CI 1.03–1.47) were associated with higher risk of grade 3–5 adverse events.

Among participants receiving the control regimen, univariate logistic regression found female sex, higher BMI, higher baseline Xpert MTB/RIF cycle threshold, history of diabetes, ethambutol $AUC_{0-24h}$, and pyrazinamide $C_{max}$ to be associated with increased risk of any grade 3–5 adverse events (threshold $P < 0.05$, Table S12). Multivariable analysis found the following factors to be associated with increased risk: female sex (OR 1.74, 95% CI 1.17–2.56) and Xpert MTB/RIF cycle threshold (OR 1.22 for every 3-cycle-threshold increase, 95% CI

1.05–1.42). Univariate and multivariable safety analyses of rifapentine-regimen and sensitivity analyses of imputed pharmacokinetic values can be found in Tables S13–15.

## External validation in RIFASHORT

We adjusted the risk stratification algorithm for use in a future clinical trial design testing only two risk groups. Consequently, the easier-to-treat TB and moderately-harder-to-treat TB phenotypes described earlier have been combined for this validation. The adjusted risk stratification algorithm included age, weight, disease extent on baseline chest radiograph, and baseline Xpert MTB/RIF cycle threshold value.

We stratified the modified intention-to-treat population across the three RIFASHORT regimens, the control 6-month 600 mg rifampin control regimen, the two 4-month high dose rifampin regimens (1200 and 1800 mg). The control regimen had 65 (31.5%) participants with the harder-to-treat TB phenotype of which 5 (7.69%) had TB-related unfavorable outcomes, and 141 (68.5%) participants in the combined easier-to-treat TB and moderately-harder-to-treat TB phenotype of which 2 (1.42%) had a TB-related unfavorable outcome. In the four-month high dose rifampin regimens, 111 (27.3%) were stratified into the harder-to-treat TB phenotype of which 15 (13.5%) had TB-related unfavorable outcomes, and 295 (72.7%) participants were in the combined easier-to-treat TB and moderately-harder-to-treat TB phenotype

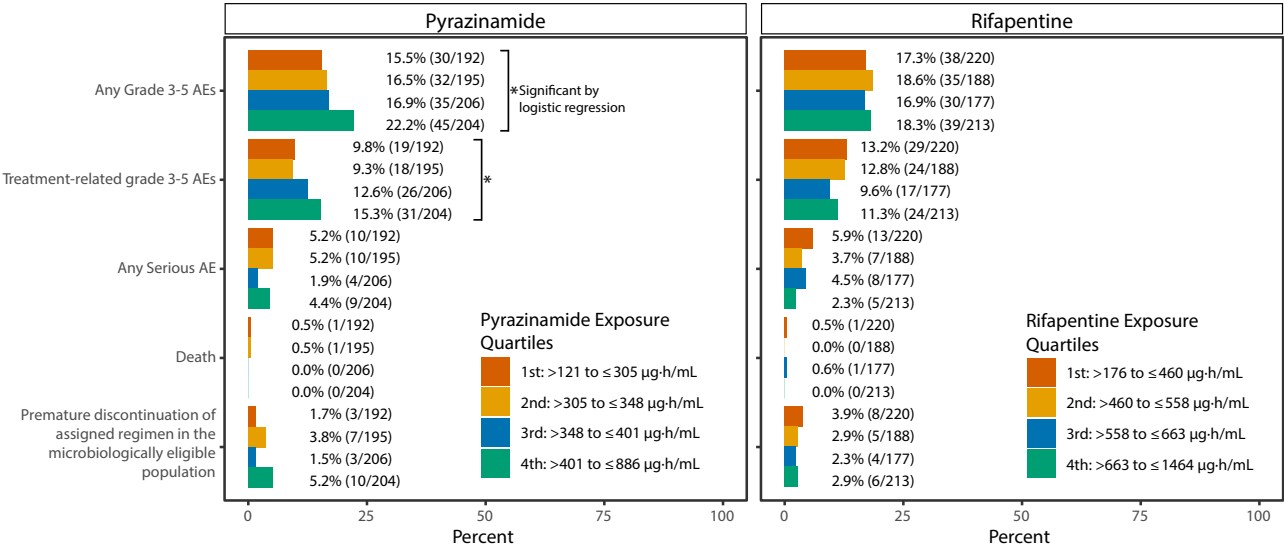

**Fig. 4 | Safety of the Rifapentine-Moxifloxacin regimen by Pyrazinamide and Rifapentine exposure.** (*) indicates significant by two-tailed logistic regression (*P* < 0.05). Among participants receiving the rifapentine-moxifloxacin regimen, higher pyrazinamide exposures were associated with increased risk of any grade 3-5 adverse events (OR 1.22 for every 100 μg•h/mL increase in $AUC_{0-24h}$, 95% CI 1.02–1.45) and treatment-related grade 3-5 adverse events (OR 1.27 for every 100 μg•h/mL increase in $AUC_{0-24h}$, 95% CI 1.04–1.55). There was no significant difference between quartiles of rifapentine exposure and any grade 3-5 adverse

events, treatment related grade 3-5 adverse events, any serious adverse events, death, or tolerability. Participants without pharmacokinetic sampling were excluded from this figure. Percentages were calculated from the safety population for all safety outcomes except for premature discontinuation of the assigned regimen, which was calculated from the microbiologically eligible population with the exclusion of participants without PK sampling. For each quartile, the percentage of participants with safety outcomes are reported with number of events in parentheses.

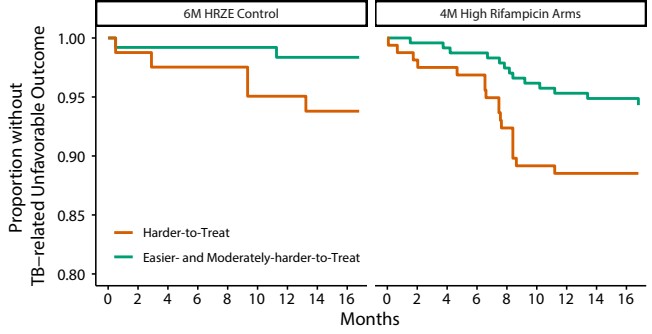

**Fig. 5 | Kaplan Meier Estimates of RIFASHORT Stratified by Treatment Duration and Risk Phenotype.** For external validation, we applied an adjusted risk stratification algorithm to the RIFASHORT modified intention-to-treat population. The easier-to-treat TB and moderately-harder-to-treat TB phenotypes were combined for this validation. The separation in the two risk groups is very clear in both the 6 M HRZE control and the 4 M high dose rifampin regimens.

of which 16 (5.42%) had a TB-related unfavorable outcome. See Fig. 1 for Kaplan Meier estimates stratified by treatment arm and risk group for visualization of the separation between the risk strata. See Fig. 5 for Kaplan Meier estimates stratified by treatment arm and risk group for visualization of the separation between the risk strata.

## Discussion

In this work, we have shown that baseline disease severity (defined by lower Xpert MTB/RIF cycle threshold and greater extent of disease on chest radiography) were risk factors for tuberculosis-related unfavorable outcomes in the 4-month experimental arms of Study 31/A5349. Low rifapentine exposure was a stronger predictor of tuberculosis-related unfavorable outcome, even after adjusting for baseline risk factors. Using just simple measures obtained at baseline, we could classify patients into disease phenotypes associated with tuberculosis-

related unfavorable outcomes. Seventy-six percent of trial participants had easier-to-treat TB (23%) or moderately-harder-to-treat TB (53%), for whom the risk of tuberculosis-related unfavorable outcome was low in the rifapentine-moxifloxacin and control regimens, indicating an opportunity for exploring further treatment shortening.

Although we found rifapentine exposure to be the primary driver of treatment success in Study 31/A5349, high rifapentine exposure was not sufficient to achieve noninferior outcomes: the substitution of ethambutol with moxifloxacin was necessary. The RIFASHORT trial tested two 4-month high-dose rifampin regimens that failed to demonstrate noninferiority compared to the 6-month standard dose rifampin regimen, which is consistent with the finding from Study 31/A5349 that the rifapentine regimen was not noninferior to the control[11]. However, while the rifapentine-moxifloxacin regimen demonstrated noninferiority, there is still room for improvement. To ensure adequate exposure in the absence of therapeutic drug monitoring, higher doses of rifapentine or longer treatment durations (although untested) would likely lead to better outcomes in the subpopulation with harder-to-treat TB[8]. Broader availability of therapeutic drug monitoring, including studies of its implementation, would also support dose adjustments; this would be beneficial since individual rifapentine exposure is highly variable (Fig. S8)[12–14].

We found no evidence that higher rifampin exposures, among participants receiving the standard dose in the control arm, decreased the risk of TB-related unfavorable outcomes. There is strong evidence in the literature that high-dose rifampin decreases time to culture conversion[15–18]; however, the lack of exposure-response for rifampin at standard dose with respect to treatment outcomes is consistent with previous studies[17,18]. Additionally, we observed only 24 (3.1%) TB-related unfavorable outcomes in the control regimen which afforded us little statistical power to detect risk factors. Study 31/A5349 availed of excellent participant adherence.

We analyzed adherence separately after considering PK and baseline factors; on-treatment factors were initially excluded to identify baseline risk factors clinicians can use to evaluate patients prior to treatment selection and initiation. Adherence was excellent in Study

31, as large majorities of participants were administered 95% of planned doses, which left little data to evaluate the adherence relationship. Nevertheless, we confirmed previous TB-ReFLECT findings that increasing adherence to treatment is also one of the most important factors in determining treatment success[6]. We found similar adjusted hazard ratios for missing one week of doses between the rifapentine-moxifloxacin and control regimens, the rifapentine-moxifloxacin regimen is more forgiving than the control, since one week of doses is 5.9% of the 4-month regimen, but 3.8% of the 6-month regimen. In multivariable analyses, rifapentine exposure remained strongly associated with unfavorable outcomes after adjusting for adherence, suggesting that the two are independent measures. As is typical in PK studies, the three doses prior to PK sampling have extra measures in place to ensure that they are administered and recorded properly, so adherence does not readily affect steady state drug exposures. Therefore, adherence measures when and for how long patients are on drug while steady state drug exposures represent the level of drug exposure achieved while on drug, both of which are extremely important to treatment success.

In Study 31, there was no clinical evidence that high rifapentine exposure or high moxifloxacin exposure were associated with an increase in adverse events or intolerability[19,20]. In contrast, higher pyrazinamide exposures in participants receiving the rifapentine-moxifloxacin regimen (multivariable) were associated with an increased incidence of grade 3–5 adverse events. Neutropenia, peripheral neuropathy, and drug-induced hepatitis have been previously reported as dose-dependent toxicities for rifapentine, moxifloxacin, and pyrazinamide, respectively. Therefore, more detailed analyses by specific adverse events are warranted to further characterize the drug-specific toxicity relationships (or lack thereof) found here.,

Our findings have implications for the design of future tuberculosis treatment trials. Previous analyses have identified easier-to-treat TB for which shorter treatments may be possible and harder-to-treat TB where large differences in treatment response between experimental and control regimens are observed[6,21]. Our analysis confirmed these findings in the rifapentine regimen, but the previous stratification based on smear grade and cavitation did not have the resolution to identify harder-to-treat TB in the rifapentine-moxifloxacin regimen. We updated the stratification algorithm with a modern measure of baseline disease burden, Xpert MTB/RIF, which other studies have confirmed is better able to discriminate between risk strata[22]. We have additionally externally validated our novel risk phenotypes with data from the recent RIFASHORT trial which tested shortened high-dose rifampin regimens. The harder-to-treat phenotype had higher incidence of TB-related unfavorable outcomes across all RIFASHORT regimens except the 1800 mg rifampin regimen which was only modestly higher (8.2% to 6.4%). The external validation demonstrates the robustness of the novel harder-to-treat phenotype definition and its potential to be applied in future trials and clinical practice. Furthermore, despite careful dose-ranging trials informing the design of the Study 31/A5349[23,24], many participants nevertheless experienced suboptimal rifapentine exposures. Collection of pharmacokinetic samples and prespecified comprehensive pharmacokinetic-pharmacodynamic analyses in Phase 3 trials is immensely valuable and can further provide critical information that guides clinical use of new regimens.

Many prediction models have been previously published and developed to predict TB treatment outcomes[25,26]. Our results are consistent with what has been observed and reported before: older age, higher weight or BMI, HIV co-infection, diabetes, male sex, and more severe baseline disease burden are risk factors for TB relapse or treatment failure. Nonetheless, our novel integrated analysis is the first to include pharmacokinetic data and more contemporary measures of baseline disease burden, both of which proved highly informative and were crucial to our understanding of treatment outcomes.

Our study has limitations. First, since the drugs were all tested as combination regimens, we could not distinguish relative contributions of each individual drug aside from comparing moxifloxacin versus ethambutol across the experimental arms. Second, we acknowledge the risks involved with subgroup analyses in trials with a noninferiority design[27], and whereas exploration of risk factors was prespecified in the parent protocol, the definitions of the three disease phenotypes presented here were not. We did, however, assess the Imperial et al. prespecified disease phenotypes[6] and validate them with the rifapentine regimen. A clinical trial being undertaken by the ACTG (SPECTRA-TB) incorporates these stratified medicine principles in the evaluation of dose-optimized rifapentine and moxifloxacin-containing ultra-short regimens. The design of that trial will provide adequate power for prespecified trial-level and stratum-level testing. Third, Study 31/A5349 was an open label trial, therefore potential biases may be present in qualitative adverse event reporting. Fourth, we chose a p-value cutoff of 0.05; while it is a reasonable cutoff selection which allowed us to identify important pharmacologically consistent risk factors, it is not the most stringent considering the large sample sizes and number of covariates and we did not use any formal statistical methods of adjustment for multiple comparisons but rather let the results speak for themselves (which were consistent with other published studies). Fifth, to preclude the use of therapeutic drug monitoring, PK was not included in the risk stratification algorithm despite its strength as a predictive risk factor. Instead, we presented the interplay between PK and risk strata for clinicians to understand the differing impacts of PK in each of the risk strata. Finally, the Study 31/ A5349 eligibility criterion of positive smear or the equivalent, as assessed by Xpert MTB/RIF, skewed the study population towards more severe pulmonary tuberculosis. Therefore, our findings do not directly address patients with sputum smear-negative pulmonary tuberculosis, estimated to account for about 40–50% of pulmonary tuberculosis cases[28,29].

In our integrated analysis of PK, demographic, and clinical factors, we have demonstrated the importance of achieving a high rifapentine exposure in the 4-month rifapentine and rifapentine-moxifloxacin regimens which reduced the risk of tuberculosis-related unfavorable outcomes, especially in individuals with more severe pulmonary tuberculosis. Furthermore, patients can be stratified by baseline disease burden into easier-to-treat TB in which further treatment shortening and simplification are likely possible and harder-to-treat TB in which longer treatment may be needed.

## Methods

### Trial design and participants

Study 31/A5349 (NCT02410772) was conducted by the Tuberculosis Trials Consortium and the AIDS Clinical Trials Group, and was funded by CDC and the National Institute of Allergy and Infectious Diseases[1,2]. Participants were ≥12 years old and had newly diagnosed pulmonary tuberculosis that was confirmed on culture to be susceptible to isoniazid, rifamycins, and fluoroquinolones[1]. The full trial protocol was published in *Contemporary Clinical Trials*[1] and was approved by the institutional review board at the U.S. CDC. An institutional review board or ethics committee at each participating trial site reviewed and approved the protocol and informed consent documents, or a trial site relied formally on the approval from the CDC. The RIFASHORT trial protocol was approved by the London School of Hygiene and Tropical Medicine Research Ethics Committee, as well as institutional and national ethics and regulatory authorities representing all participating sites and countries. All the participants provided written informed consent.

### Pharmacokinetics

All randomized participants underwent steady state pharmacokinetic sampling between weeks 2–8 of treatment. Intensive sampling

was performed on a minority of participants with samples taken at 0.5, 3, 5, 9, 12, and 24 hours after ingestion of the reference dose. The remaining participants were sampled sparsely with time points at 0.5, 5, and 24 hours. Plasma concentrations of all drugs were determined using validated high-performance liquid chromatography mass spectroscopy assays. Population pharmacokinetic models were developed for each drug, and individual area under the concentration-time curve from 0–24 hours ($AUC_{0-24h}$) and maximal plasma concentration ($C_{max}$) were calculated (Chang V, Imperial MZ, Zhang N, Phillips PPJ, Nahid, P, Dorman SE, Weiner M, Kurbatova EK, Whitworth WC, Bryant KE, Carr W, Engle ML, Nhung NV, Nsubuga P, Diacon A, Dooley KE, Chaisson RE, Swindells S, Savic RM, Rifapentine Population Pharmacokinetics and Dosing Recommendations for the Treatment of Tuberculosis from a Phase 3 Confirmatory Trial [Manuscript submitted for publication]). $AUC_{0-24h}$ and $C_{max}$ were imputed for participants with missing pharmacokinetic samples using the population pharmacokinetic models [30].

### Liquid Chromatography Mass Spectroscopy assays
5028 samples were analyzed at atlanbio for rifapentine and 2377 for moxifloxacin. Sample integrity will be verified upon reception and samples will be stored at approximately −80 °C. The LC-MS/MS analysis will be carried out with: (1) shimadzu liquid chromatography system and autosampler, (2) an analytical chromatographic column, and (3) a triple quadripole mass spectrometer system working in the heated electron spray ionization positive mode. Software used included Analyst (for moxifloxacin) and LCQuan (for rifapentine and 25-desacetyl rifapentine) for LC-MS/MS instrument control, data acquisition and chromatogram peak integration. Watson® LIMS software (Thermo Electron, Philadelphia, PA) was used for sample management and data management including regression, concentration calculations, statistics. For each method and each batch of analysis, unless the method is already running, the performance will be verified before the start of the analysis of the study samples. The set-up run will include a calibration curve and quality control at three levels: QC Low, QC Medium and QC High (6 replicates per level).

### Batch Acceptance criteria
- Deviation for calibration standards should be within ± 15% from nominal concentrations, except for the LLOQ for which it should be within ± 20%. In case a standard does not comply with these criteria, it will be rejected, and the calibration curve without this standard will be re-evaluated.
- At least 75% of calibration standards should meet the above criteria, with at least six concentration levels.
- At least 2/3 of all QC should be within 15% of their nominal value, with at least 50% of QC meeting acceptance criteria at each level.
- QC0 should be BLOQ (except if one-off contamination has been evidenced).

### Demographics and clinical factors
We considered the following baseline factors potentially associated with treatment efficacy: age, sex, self-reported race, trial site, weight, body-mass index (BMI), Karnofsky performance scale score, HIV status, diabetes history, and smoking history. Baseline sputum measurements included acid-fast bacillus smear grade, time-to-positivity in liquid culture (Mycobacteria Growth Indicator Tube, Becton Dickinson), and Xpert MTB/RIF cycle threshold (Xpert MTB/RIF, Cepheid). Sex was self-reported, if participants did not want to answer "unknown" was recorded. Chest radiography measurements considered included cavitary disease, aggregate cavity size, and extent of disease defined as percent involvement of the thoracic area. Finally, we considered adherence measured as total number of doses taken as an on-treatment factor potentially associated with efficacy. Participants

with missing baseline covariates were excluded from multivariable analyses.

### Efficacy outcomes
The efficacy outcome in the present analysis was time to tuberculosis-related unfavorable outcome within one year post-treatment initiation[1,2]. Tuberculosis-related unfavorable outcomes were defined as: (1) two consecutive positive sputum cultures on or after week 17, (2) not seen at month 12 with last culture positive, or (3) clinical diagnosis of tuberculosis recurrence and treatment restarted. Tuberculosis-unrelated unfavorable outcomes and not assessable outcomes (e.g., participants not seen at month 12 with a negative last culture or withdrawn due to pregnancy) were right-censored at the time of visit that led to that status; favorable outcomes were right-censored at the time of last follow-up visit.

### Safety outcomes
The primary safety outcome was the occurrence of any grade 3–5 adverse event[31]. We also considered the following predefined safety outcomes: treatment-related grade 3–5 adverse events, serious adverse events, death, and premature discontinuation of the assigned treatment for any reason other than microbiological ineligibility.

### Statistical analysis
We generated Kaplan-Meier estimates, stratified by $AUC_{0-24h}$ dichotomized at the population median, and performed pharmacokinetic-pharmacodynamic Cox proportional hazards analysis for each of the six drugs and each arm separately.

The analysis population consisted of the microbiologically eligible population[2]. We evaluated the proportional hazards assumption for all demographics, baseline clinical factors, and continuous pharmacokinetic parameters, then tested each in univariate and multivariable Cox analyses for each regimen separately to identify risk factors for tuberculosis-related unfavorable outcomes. We selected risk factors for the final multivariable model with a stepwise procedure, testing linear relationships in a forward inclusion and backwards exclusion procedure (likelihood ratio test $P < 0.05$). The selected risk factors were used to construct a risk stratification algorithm that stratified disease phenotypes into easier-to-treat TB, moderately-harder-to-treat TB, or harder-to-treat TB. For each risk stratum, we performed subgroup analyses calculating the risk difference and 95% Wald confidence interval comparing each experimental arm to the control; we compared the upper border of the confidence interval to a 6.6% margin, the threshold for noninferiority used in the primary analysis[2,32,33]. We also tested prespecified TB-ReFLECT disease phenotype definitions from Imperial et al[6]. For external validation of the risk stratification findings, we applied an adjusted risk stratification algorithm to the RIFASHORT (NCT02581527) patient population [11].

We used logistic regression to evaluate the association between $AUC_{0-24h}$ and $C_{max}$ of all drugs and safety outcomes. We considered demographics, baseline clinical factors, and pharmacokinetic parameters as potential predictors of any grade 3–5 adverse events in univariate and multivariable logistic regression. The selection of covariates followed the same stepwise procedure described above.

### Reporting summary
Further information on research design is available in the Nature Portfolio Reporting Summary linked to this article.

## Data availability
The CDC is currently preparing de-identified TBTC Study 31/A5349 patient data to be made available via a recognized data sharing platform. De-identified TB-ReFLECT data was received from TB-PACTS and access can be requested here (https://c-path.org/tools-platforms/tb-pacts/).

## Code availability

R scripts for main figures is made available in the supplementary. NONMEM control streams are available upon request. No custom packages were used for this analysis.

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

## Acknowledgements

We especially thank the study participants and the local Tuberculosis program staff who contributed their time to Study 31/A5349 and the RIFASHORT trial. We thank our IND sponsor, the U.S. Centers for Disease Control. We thank the AIDS Clinical Trial Group for their vast network of clinical trial sites dedicated to improving HIV care and innovating TB treatments. We thank Doctors Phil LoBue, Carla Winston, and Jonathan Mermin for continued support of the Tuberculosis Trials Consortium within the CDC; Westat, Inc. and PPD, Inc. for on-site monitoring. We thank Daniel Grint and the RIFASHORT team for providing us access to the data for external validation. We are grateful to Richard Hafner and Peter Kim, who provided support for PK/PD work, to Stefan Goldberg and Nigel Scott for assistance with study drug management, to Anne Purfield, Nicole Brown, and Jessica Ricaldi for support to the study mycobacteriology laboratories. Sanofi (Paris, France, and Bridgewater, New Jersey, USA) donated rifapentine and all other study drugs, supported shipping of study drugs to all sites, and provided funding support for pharmacokinetic testing and preparation of the final Clinical Study Report.

Funding support for this trial was provided by the U.S. Centers for Disease Control and Prevention, National Center for HIV/AIDS, Viral Hepatitis, STD, and TB Prevention, Division of Tuberculosis Elimination (contracts 200-2009-32582, 200-2009-32593, 200-2009-32594, 200-2009-32589, 200-2009-32597, 200-2009-32598, 75D30119C06702, 75D30119C06701, 75D30119C06703,

75D30119C06222, 75D30119C06225, 75D30119C06010); and by the National Institute of Allergy and Infectious Diseases (NIAID) of the National Institutes of Health under Award Numbers UM1 AI068634, UM1 AI068636 and UM1 AI106701. GEV is supported by NIAID K08 AI141740. KED is supported by NIAID K24 AI150349. RMS is supported by NIH/NIAID R01 AI135124-01A1.

## Author contributions

Study conception and design: R.M.S., P.P.J.P, P.N. Data collection: P.Nahid, A.V., E.V.K., S.S., R.E.C., S.E.D., J.L.J., M.W., E.E.S., W.W., W.C., K.E.B., D.B., M.E., P. Nsubuga, A.H.D., N.V.N., R.D., A.J., T.S.H. Data analysis: V.K.C., M.Z.I., P.P.J.P., R.M.S. Data interpretation: V.K.C., M.Z.I., P.P.J.P., R.M.S., P.Nahid, G.E.V. Drafting of the initial manuscript: V.K.C., P.P.J.P., R.M.S., P.Nahid, G.E.V., A.V., S.S., R.E.C., S.E.D., W.C., K.E.D. Critical Review of the Final Draft of the Manuscript: V.K.C., M.Z.I., P.P.J.P., R.M.S., P.Nahid, G.E.V., A.V., E.V.K., S.S., R.E.C., S.E.D., W.C., K.E.D., J.L.J., E.E.S., K.E.B., M.E., P.Nsubuga, A.H.D., N.V.N., R.D. Access and Verification of Underlying Data: K.E.B., W.W., V.K.C., E.V.K.

## Competing interests

The authorship team members have declared no potential conflicts of interest with respect to the research, authorship, or publication of this article. Sanofi commercial interests did not influence the study design; the collection, analysis, or interpretation of data; the preparation of this manuscript; or the decision to submit this manuscript for publication. A Sanofi technical expert served on the protocol team.

## Additional information

[1]Department of Bioengineering and Therapeutic Sciences, University of California San Francisco, San Francisco, CA, USA. [2]UCSF Center for Tuberculosis, University of California San Francisco, San Francisco, CA, USA. [3]Division of Pulmonary and Critical Care Medicine, University of California San Francisco, San Francisco, CA, USA. [4]Division of HIV, Infectious Diseases, and Global Medicine, University of California San Francisco, San Francisco, CA, USA. [5]Centers for Disease Control and Prevention, Atlanta, GA, USA. [6]University of Nebraska Medical Center, Omaha, NE, USA. [7]Center for Tuberculosis Research, Johns Hopkins University School of Medicine, Baltimore, MA, USA. [8]Medical University of South Carolina, Charleston, SC, USA. [9]Case Western Reserve University, University Hospitals Cleveland Medical Center, Cleveland, OH, USA. [10]Uganda–Case Western Reserve University Research Collaboration, Kampala, Uganda. [11]University of Texas Health Science Center at San Antonio and the South Texas Veterans Health Care System, San Antonio, TX, USA. [12]St. George's, University of London, London, UK. [13]Division of Infectious Diseases, Vanderbilt University, Nashville, TN, USA. [14]TASK, Cape Town, Western Cape, Cape Town, South Africa. [15]Vietnam National Tuberculosis Program–University of California, San Francisco Research Collaboration Unit, Hanoi, Vietnam. [16]National Lung Hospital, Hanoi, Vietnam. [17]Lung Institute and Division of Pulmonology, Department of Medicine, University of Cape Town, Cape Town, South Africa. ✉e-mail: rada.savic@ucsf.edu

## AIDS Clinical Trial Group

Harriet Mayanja Kizza[10], Pheona Nsubuga[10], Elias Ssaku[10], Isaac Sekitoleko[10], Joseph P. Akol[10], Andreas Diacon[14], Carmen Kleinhans[14], Julia Sims[14], Rodney Dawson[17], Erika Mitchell[17], Bronwyn Hendricks[17], Yvetot Joseph[18], Marie Jude Jean Louis[18], Cadette Mercy[18], Alexandra Apollon[18], Gertrude Royal[18], Pamela Mukwekwerere[19], Yeukai Musodza[19], Wilfred Gurupira[19], Michele Tameris[20], Angelique Kany Kany Luabeya[20], Mark Hatherill[20], Mario Camblart[21], Circée Phara Jean[21], Mohammed Rassool[22], Noluthando Mwelase[22], Jaclyn Bennet[22], Lerato Mohapi[23], Ntebo Mogashoa[23], Debra Peters[23], Sanjay Gaikwad[24], Neetal Neverkar[24], Rahul Lokhande[24], Cornelius Munyanga[25], Mina Hosseinipour[25], Charity Potani[25], Elisha Okeyo[26], Samuel Gurrion Ouma[26], Prisca Rabuogi[26], Rodrigo Escada[27], Lidiane Tuler[27], Johnstone Kumwenda[28], Kelvin Mponda, Lynne Cornelissen[29], Andriette Hiemstra[29], Umesh G. Lalloo[30], Sandy Pillay[30], Abraham Siika[31], Alberto Mendoza[32], Pedro Gonzales[32], Mey Leon[33], Javier R. Lama[33], Alvaro Schwalb[34], Eduardo Gotuzzo[34], Fredrick Sawe[35], Isaac Tsikhutsu[35], Sivaporn Gatechompol[36], Anchalee Avihingsanon[36], Natthapol Kosashunhanan[37], Patcharaphan Sugandhavesa[37], Marineide Gonçalves de Melo[38], Rita de Cassia Alves Lira[38], Anne Luetkemeyer[2], Carina Marquez[2], Kristine Coughlin[39], Kelly E. Dooley[7], Jacques H. Grosset[7], Eric L. Nuermberger[7], Lara Hosey[40], Anthony T. Podany[6], Andrey Borisov[5], Nicole Brown[5], Deron Burton[5], Scott Burns[5], Wendy Carr[5], Crystal Carter[5], Lauren Cowan[5], Melinda Dunn[5], Barbara DeCausey[5], Melissa Fagley[5], Kimberly Hedges[5],

**Constance Henderson**[5], **Amanda Hott**[5], **Carla Jeffries**[5], **Katherine Klein**[5], **Joan Mangan**[5], **Gerald Mazurek**[5], **Ruth Moro**[5], **Lakshmi Peddareddy**[5], **James Posey**[5], **Mary Reichler**[5], **Jessica Ricaldi**[5], **Claire Sadowski**[5], **William Whitworth**[5], **Melisa Willby**[5], **Yan Yuan**[5] **& April C. Pettit**[41]

[18]Les Centre GHESKIO INLR, Port au Prince, Port au Prince, Haiti. [19]Parirenyatwa Clinical Research Site, Harare, Zimbabwe. [20]South African Tuberculosis Vaccine Initiative (SATVI), Worcester, South Africa. [21]Les Centre GHESKIO IMIS, Port au Prince, Haiti. [22]Wits Helen Joseph Clinical Research Site Department of Medicine, Johannesburg, South Africa. [23]Soweto ACTG Clinical Research Site, Soweto, South Africa. [24]Byramjee Jeejeebhoy Medical College, Pune, India. [25]University of North Carolina Project Tidziwe Centre, Lilongwe, Malawi. [26]Kisumu Clinical Research Site, Kisumu, Kenya. [27]Institute de Pesquica Evandro Chagas Fiocruz, Rio de Janeiro, Brazil. [28]Blantyre Clinical Research Site, Johns Hopkins Research Project, Blantyre, Malawi. [29]Family Clinical Research Unit, University of Stellenbosch, Parow Valley, South Africa. [30]Durban International Clinical Research Site, Durban, South Africa. [31]Moi University Clinical Research Site, Eldoret, Kenya. [32]San Miguel Clinical Research Site, IMPACTAPERU, Putamayo, Peru. [33]Asociacion Civil Impacta Salud y Educacion, Lima, Peru. [34]Universidad Peruana Cayetano Heredia, San Martín de Porres, Lima, Peru. [35]Kenya Medical Research Institute/Walter Reed Project Clinical Research Center, Kericho, Kenya. [36]The Thai Red Cross AIDS Research Centre, Bangkok, Thailand. [37]HIV Treatment Clinical Research Site, Chiang Mai University, Chiang Mai, Thailand. [38]Hospital Conceicao, Porto Alegre, Brazil. [39]Frontier Sciences, Amherst, New York, NY, USA. [40]Social and Scientific Systems, AIDS Clinical Trials Group Operation Center, New York, NY, USA. [41]Vanderbilt University School of Medicine, Nashville, TN, USA.

## Tuberculosis Trials Consortium

**Lien T. Luu**[15], **Hanh T. T. Nguyen**[15], **Hung V. Nguyen**[15], **Hue T. M. Nguyen**[16], **Cyndy Merrifield**[15], **Matebogo Xaba**[42], **Maya Jaffer**[42], **Keitumetse Majoro**[42], **Kwok-Chiu Chang**[43], **Chi Chiu Leung**[43], **Polo Pavon**[11], **Rogelio Duque Jr**[11], **George Samuel**[44], **Joseph Burzynski**[45], **Mascha Elskamp**[45], **Jill Campbell**[46], **Marlon Quintero**[46] **& Elizabeth Guy**[47]

[42]Wits Health Consortium Perinatal HIV Research Unit (PHRU), Johannesburg, South Africa. [43]Tuberculosis and Chest Service of Hong Kong, Hong Kong, China. [44]University of North Texas Health Science Center, Fort Worth, TX, USA. [45]Columbia University, New York, NY, USA. [46]Austin Tuberculosis Clinic, Austin, TX, USA. [47]Baylor College of Medicine & Affiliated Hospitals/VA, Houston, TX, USA.

