## [Peer Review File · Nature Communications]

Reviewer #2 Comments:

Most of my comments were addressed properly, with the addition of external and internal validity.

For my previous comment 7, "2) The impact of Rifapentine exposure was not included in the model, while this may highly relate to the outcomes, which may induce bias, as Low rifapentine exposure was a stronger predictor of tuberculosis-related unfavorable outcome." I understand that including this would require therapeutic drug monitoring and would preclude the use of the risk stratification algorithm at baseline. But without considering this, it may still induce bias, and the authors should mention this in the limitation section.

Reviewer #3 Comments:

The manuscript has been revised, with the most substantial revision being external validation using data from two other trials, one of which they are allowed to include in the paper. The revised report does strongly suggest that risk stratification using disease extent on CXR and Xpert ct values can identify patients at low risk of worse outcomes when treated with a 4-month regimen. This finding is very similar to the group's previous studies (Imperial et al, Nat Med 2018, and Imperial et al, AJRCCM 2021) showing that AFB smear grade and cavitation or a six-feature list of characteristics including AFB smear and cavitation could be used to identify patients whose treatment response rates using 4-month regimens were non-inferior to control regimen rates. Therefore the primary point of the article - that low risk groups eligible for 4-month regimens exist and can be readily identified - has already been made conceptually in two prior publications. The main new data here, in terms of predictions, relates to the use of Xpert ct value rather than AFB smear grade, which the authors state is a more modern approach to evaluating disease burden. I agree with this assertion, but conceptually both AFB smear and Xpert ct values are measuring pathogen burden, so this advance is incremental and not conceptual. While the PK data are presented, the authors choose to not incorporate these data into the risk stratification algorithms, such that the end result is another risk stratification approach highly similar to approaches they have previously reported. If the point of this article is primarily risk stratification, and if PK data aren't going to be used to stratify patients, it isn't clear why PK data are incorporated into the algorithms (Figure 2) or why substantial text is dedicated to PK as a risk factor.

Reviewer #2 Comments:

Most of my comments were addressed properly, with the addition of external and internal validity.

For my previous comment 7, "2) The impact of Rifapentine exposure was not included in the model, while this may highly relate to the outcomes, which may induce bias, as Low rifapentine exposure was a stronger predictor of tuberculosis-related unfavorable outcome. " I understand that including this would require therapeutic drug monitoring and would preclude the use of the risk stratification algorithm at baseline. But without considering this, it may still induce bias, and the authors should mention this in the limitation section.

Thank you for the comment, we added the following sentences into the limitations section of the discussion lines 358-360.

"Fifth, to preclude the use of therapeutic drug monitoring, PK was not included in the risk stratification algorithm despite its strength as a predictive risk factor. Instead, we presented the interplay between PK and risk strata for clinicians to understand the differing impacts of PK in each of the risk strata."

Reviewer #3 Comments:

The manuscript has been revised, with the most substantial revision being external validation using data from two other trials, one of which they are allowed to include in the paper. The revised report does strongly suggest that risk stratification using disease extent on CXR and Xpert ct values can identify patients at low risk of worse outcomes when treated with a 4-month regimen. This finding is very similar to the group's previous studies (Imperial et al, Nat Med 2018, and Imperial et al, AJRCCM 2021) showing that AFB smear grade and cavitation or a six-feature list of characteristics including AFB smear and cavitation could be used to identify patients whose treatment response rates using 4-month regimens were non-inferior to control regimen rates. Therefore the primary point of the article - that low risk groups eligible for 4-month regimens exist and can be readily identified - has already been made conceptually in two prior publications. The main new data here, in terms of predictions, relates to the use of Xpert ct value rather than AFB smear grade, which the authors state is a more modern approach to evaluating disease burden. I agree with this assertion, but conceptually both AFB smear and Xpert ct values are measuring pathogen burden, so this advance is incremental and not conceptual. While the PK data are presented, the authors choose to not incorporate these data into the risk stratification algorithms, such that the end result is another risk stratification approach highly similar to approaches they have previously reported. If the point of this article is primarily risk stratification, and if PK data aren't going to be used to stratify patients, it isn't clear why PK data are incorporated into the algorithms (Figure 2) or why substantial text is dedicated to PK as a risk factor.

This point is well made, and one we debated endlessly in the crafting of this manuscript. Ultimately, we decided to only include baseline risk factors into the algorithm to preclude the need of therapeutic drug monitoring to use this updated risk algorithm. We agree that therapeutic drug monitoring could be a useful tool given the variability of rifapentine exposure and so this is another reason why we do include the PK models. We note this in lines 201-203 in

the results section and lines 287-290 in the discussion. We have also added new text into the limitations section highlighting this point, lines 358-360. Despite the non-inclusion of PK into the risk algorithm, it remains an important risk factor to understand. Thus, in figure 2 we stratified by baseline risk factors and PK, so that the interplay of PK and baseline risk factors can be understood. In figure 2, we see that achieving higher rifapentine exposures has a larger impact in the harder-to-treat phenotype compared to the easier-to-treat phenotype. Even if PK is not included into the risk stratification, this knowledge is important to clinicians, who may opt to increase the dose of rifapentine for patients who are in the harder-to-treat phenotype. However, without testing the risk stratification and dose adjustments in clinical settings, we cannot recommend changes to clinical practice and can only suggest that they be researched further.

REVIEWERS' COMMENTS

Reviewer #1 (Remarks to the Author):

I reviewed the comments from previous reviewers as well as the response of the authors. The major concerns of the previous reviewers were addressed properly, and I have no further comments.